# Application Research of Improved YOLO V3 Algorithm in PCB Electronic Component Detection

**Jing Li [1,2]**, **Jinan Gu [1,*]**, **Zedong Huang [1]** and **Jia Wen [3]**

[1] School of Mechanical Engineering, Jiangsu University, Zhenjiang 212000, China
[2] School of Electronic Information and Electrical Engineering, Anyang Institute of Technology, Anyang 455000, China
[3] Frontier Information Technology Institute, Zhongyuan University of Technology, Zhengzhou 450007, China
[*] Correspondence: gujinan@tsinghua.org.cn



**Featured Application: In summary, the contribution of this paper is three-fold. First, we establish a joint training dataset for electronic components that includes real PCB photos and virtual PCB photos based on circuit simulation software. Second, we propose an improved YOLO (you only look once) V3 algorithm that adds one YOLO output layer that is sensitive to small targets and validates the effectiveness of the algorithm in a real PCB picture and virtual PCB picture test including a large number of PCB electronic components. Third, we use the clustering algorithm to design and generate 12 anchor boxes suitable for electronic components and assign these anchor boxes to the above four YOLO output layers.**

**Abstract:** Target detection of electronic components on PCB (Printed circuit board) based on vision is the core technology for 3C (Computer, Communication and Consumer Electronics) manufacturing companies to achieve quality control and intelligent assembly of robots. However, the number of electronic components on PCB is large, and the shape is different. At present, the accuracy of the algorithm for detecting all electronic components is not high. This paper proposes an improved algorithm based on YOLO (you only look once) V3 (Version 3), which uses a real PCB picture and a virtual PCB picture with synthesized data as a joint training dataset, which greatly increases the recognizability of training electronic components and provides the greatest possibility for data enhancement. After analyzing the feature distribution of the five dimensionality-reduced output layers of Darknet-53 and the size distribution of the detection target, it is proposed to adjust the original three YOLO output layers to four YOLO output layers and generate 12 anchor boxes for electronic component detection. The experimental results show that the mean average precision (mAP) of the improved YOLO V3 algorithm can achieve 93.07%.

**Keywords:** electronic component detection; PCB; improved YOLO V3; virtual image; YOLO outputs; synthesized data

---

## 1. Introduction

The electronic information industry has the characteristics of high technology content, high added value and low pollution. In addition to traditional digital cameras, mobile phones, computers, and other 3C (Computer, Communication and Consumer Electronics) products, emerging 3C products have emerged in recent years: smart TVs, smartwatches, smart wearable devices, drones, sweeping robots, entertainment robots, etc. With the further development of market hotspot products represented by flat-panel TVs and smartphones, the electronic information industry has become increasingly influential in social change and is regarded as a strategic development industry by major countries around the

world. The traditional 3C industry is a labor-intensive industry, and labor is one of the major costs of 3C manufacturing companies. At the same time, the new generation of labor is reluctant to engage in repetitive work that is boring with low added value, causing the 3C manufacturing industry to face a serious labor shortage. Based on the above situation, 3C terminal manufacturers must improve production automation by reducing labor shortages, improving productivity, enhancing competitive advantage, and completing industrial transformation and upgrading. That is, 3C manufacturing is shifting from labor-driven to equipment-driven.

With the advent of the industry 4.0 era, the role of machine vision in the 3C manufacturing field is becoming more and more important. According to the application function, it is mainly reflected in four aspects: measurement, detection, identification and positioning, especially PCB (Printed circuit board) electronics based on intelligent vision technology. Component detection is an important step in PCB automation production monitoring and hardware reverse engineering. PCB electronic component detection belongs to the target detection category. Target detection is usually done in two steps in the literature: (i) the possible position of the target in the video or picture of the recognition object is fed back from the camera, and (ii) classifying the signs for a given cropped image where only a sign (or maybe nothing relevant) is visible, determining whether it is a sign or not, and what sign it is [1]. That is, target detection can be understood as a combination of object recognition and object location [2,3].

Various target detection visual algorithms have been evolving over the past three decades. At present, the target detection algorithms appearing in academia and industry are mainly divided into three categories: traditional object detection algorithm, two-stage method, and one-stage method. Traditional target detection is mostly based on statistics or knowledge. Its characteristic attributes rely on manual design, and the detection objects are relatively limited. The face and license plate are the main ones. The algorithm is based on cascade + Harr/SVM (support vector machine) + HOG (histogram of oriented gradient)/DPM (deformable part model), and its improvement and optimization algorithms. Yun et al. proposed a detection algorithm based on Harr and directional gradient histogram (HOG) features. The algorithm makes full use of the HOG feature of the target vehicle and uses the Harr feature to extract the foreground area (ROI, Region of interest). The AdaBoost classifier of the joint structure performs feature and target area classification and detects and tracks multiple vehicle targets successfully in a complex urban environment [4]. Deng et al. proposed a multi-resolution layered component model and corresponding coarse-to-fine inference process which can be cascaded with the deformable part to multiply the acceleration factor to achieve accelerated object detection [5]. However, deep learning is gradually becoming the mainstream in target detection with its enormous success beginning with the ImageNet Large Scale Visual Recognition Challenge (ILSVRC) of 2012 [6]. The two-stage method is also called the deep learning classification algorithm. The algorithm first performs region proposal extraction on the detected images. Region proposal uses the texture, color and other information in the image, so the number is small, and the quality is high. It also uses the deep neural network to automatically extract and classify these candidate windows. Then it merges the regions that contain the same target, and finally outputs the target region we want to detect. Such algorithms are represented by RCNN (Regions with convolutional neural network features), Fast RCNN, Faster RCNN, SPP-net (Spatial pyramid pooling), and R-FCN (Region-based fully convolutional networks). Han et al. proposed a real-time small-flow symbol detection method based on revised Faster RCNN. Firstly, the small region proposal is used to extract the characteristics of small traffic signs, and then the revision structure of Faster RCNN combines the online hard instance mining. The combination can effectively detect small and medium traffic signs in surveillance video or driving recorder videos [7]. Ni et al. proposed a cascaded network to identify gestures in complex scenes. First, they used a region-based complete convolutional neural network (R-FCN), which detects small objects, detects gestures, and then used the online hard instance test (OHET) to select hard instances to further improve the accuracy of gesture recognition. The VGG-19 (Visual geometry group-19)classification network reclassifies hard instances to obtain the final output of the gesture recognition system; the latest results have achieved a 99.3% mean average precision (mAP) of small and similar gestures in complex

scenes [8]. The deep learning regression algorithm that is the one-stage method is represented by YOLO (you only look once), SSD (single-shot multi-box detectors), etc. This kind of target detection algorithm discards the process of generating regional proposal and directly returns the position and category of the bounding box in the output layer, which greatly improves the detection speed. Liu et al. did not need to extract features manually, using two ZiYuan3 (ZY3) remote sensing images to create six training data sets with different band combinations and sliding window sizes, and training single-shot multi-box detectors (SSD) models, respectively. This proposed method obtains a higher rate of identification of poppy plots [9]. Tang et al. proposed a multi-view YOLO object detection method, model structure and working method, which improved the ability to detect small objects. Compared with the two-stage method, this multi-view one-stage method is faster when it reaches the same map when implementing small object detection [2].

Vision-based inspection of electronic components on PCB can solve the problem of defect detection in 3C manufacturing. However, due to the wide variety and variety of electronic components, it has always been an industry problem. Zhou et al. found typical PCB electronic components, including protective coatings, markings and solder joints, proposed the detection of solder joints based on specular properties and the determined the specular interval of gray levels by the multilevel thresholding algorithm while passing the solder joint color. The distribution features are compared with the color distribution features of the reference object to identify and remove marks, via and other ineffective specular reflection regions, ultimately enabling automatic detection of multiple electronic components in the PCB image [10]. The electronic components detected by Chigateri et al. were transistors, capacitors and diodes. Therefore, a knowledge-based method is proposed for target detection, that is, first it determines the aspect ratio and the pin number of the component, then it calculates the roundness of the device, and finally achieves the detection target of the electronic component through correlation matching [11]. Li et al. used a novel background removal algorithm and by appropriately combining complementary information sources (e.g., diverse color spaces, scales, etc.), significantly improved localization performance in electronic components on PCB [12]. With the development of artificial intelligence technology, the use of deep learning to target detection of PCB electronic components can achieve the goal of robotic assembly in 3C manufacturing and production, and it has increasingly attracted the attention of researchers. Tang et al. has established a PCB defect dataset and proposed a novel group pyramid pooling module to extract features of a large range of resolutions, which were merged by group to predict PCB defect of corresponding scales [13]. Huang et al. produced and publicized a synthesized PCB dataset that has 1386 images with six types of common defects. Instead of simply superimposing the convolution layer, the authors used a dense shortcut from Densenet to achieve satisfactory defect detection with only a relatively small number of layers [14]. Kuo et al. were one of the first teams to try to detect common components on PCBs based on deep learning models, and they proposed a data set for common component inspection on PCBs and designed a novel Graph Network block to refine the component features conditioned on each PCB. Using the model SPN-T-W-GN-LF (Similarity prediction network with triplet loss and graph network), the mAP of electronic component detection on the testing of PCBs can reach 65.3% [15]. The previous two studies focused on the detection of a few types of electronic components in a simple scenario, although [15] used the embedded graph neural network to enhance feature implementation with the aim of detecting multi-target electronic components in a complex background, but since it is a three-stage pipeline, the problem of low efficiency and low robustness exists. We used the improved YOLO (you only look once) V3 [16] algorithm after the network structure changed based on the fast detection speed of a regression method to find electronic components. The detection accuracy significantly improved using virtual images for data enhancement and feature fusion based on different categories of training samples' scale. Figure 1 shows the technical flowchart of this study [17].

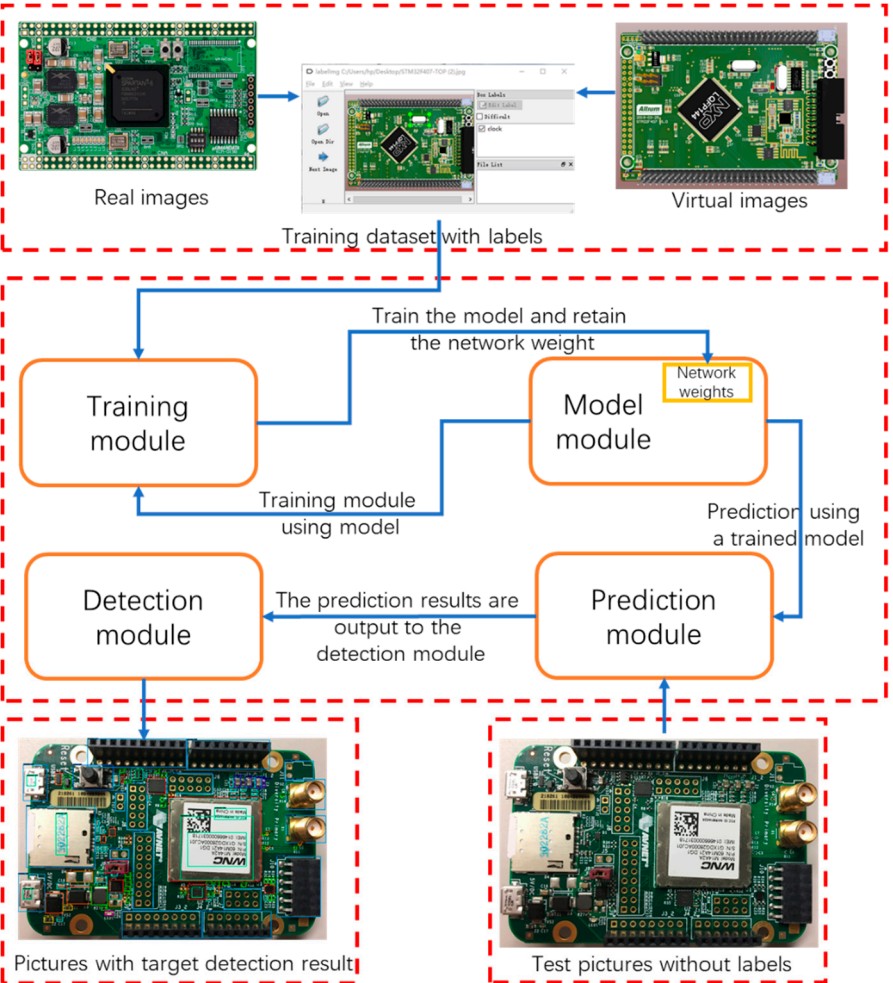

**Figure 1.** Technical flowchart of the study.

The contributions of this paper are summarized as follows:

- We propose a PCB electronic component category training dataset that involves the virtual environment and real-world scenario collaboration and validate its effectiveness in a typical multi-category test set.
- We use the K++ clustering algorithm to regenerate the anchors suitable for the distribution of PCB electronic components based on the size ratio of the target in the training dataset.
- We design an improved YOLO V3 network structure and extend the exit of the YOLO layer by extracting the characteristics of the network layer suitable for small targets. This improved network structure is particularly suitable for detecting small SMD (Surface mounted devices) components on the PCB. The improved model's mAP can reach 93.07% of the PCB electronic components after comparison with the original YOLO V3.

The rest of the paper is organized as follows: The improvements between YOLO V3 and the previous YOLO series are discussed in Section 2. Section 3 proposes an improved scheme based on YOLO V3 that can improve the target detection accuracy by analyzing the training dataset and test dataset of PCB electronic components. Section 4 provides the experimental results and analysis in detail. Section 5 concludes the paper and discusses future research issues.

## 2. Related Work

The YOLO algorithm was proposed by Redmon et al. [18] on CVPR2016 (IEEE conference on computer vision and pattern recognition of 2016). After two years of development, it has grown from YOLO V1 to YOLO V3. The YOLO algorithm is a typical one-stage target detection algorithm that combines classification and target regression problems with an anchor box, thus achieving high efficiency, flexibility and generalization performance. Its backbone network Darknet can also be replaced with many other frameworks, so it is very popular in the engineering field. We will introduce the core technology of YOLO V3 in detail. The work done in this section will be the basis for the improved algorithm in Section 3.

### 2.1. Network Architecture

YOLO V3 consists of the backbone network called Darknet-53, the upsampling network, and the detection layers called YOLO layers [19]. The backbone network of YOLO V3 is based on Darknet-53 to extract feature images from the input image. The whole network mainly uses residuals as the basic components. A total of five residual layers with different scales and depths are selected, and they only perform residuals between outputs of different layers. Each residual block consists of a pair of $3 \times 3$ and $1 \times 1$ convolutional layers with quick connections, and the total number of convolutional layers is 53. The size of the final feature map has a spatial resolution 1/32 smaller than the input image. YOLO V3 uses a three-scale YOLO layer that is responsible for detecting objects of different scales. In the first YOLO layer, the grid resolution is 1/32 of the input image detecting large objects. The resolution of the last YOLO layer is 1/8, which is capable of detecting small objects. There are multiple convolution layers and one upsampling layer between the YOLO layers. Each layer has a convolution sublayer, a batch normalization sublayer and a leaked ReLU (Rectified linear unit) activation sublayer. Some quick connections connect the middle layer of Darknet-53 to the layer after the upsampling layer. The network structure for YOLO V3 is shown in Figure 2.

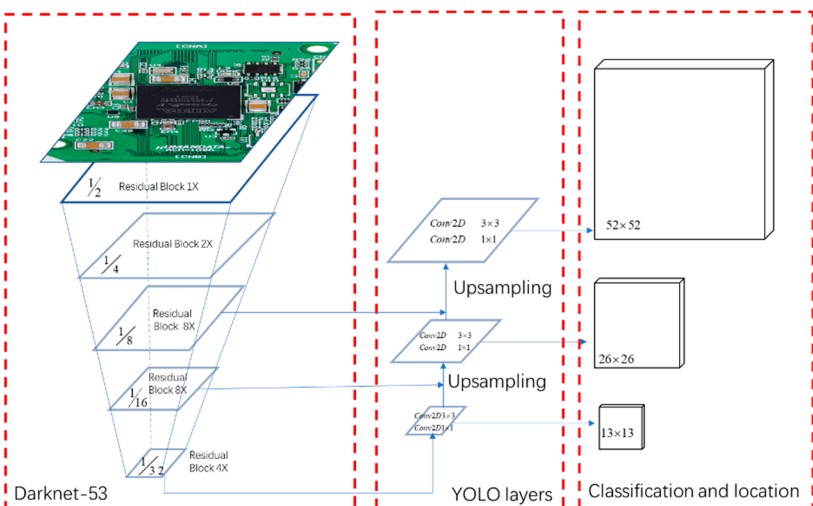

**Figure 2.** Schematic of the YOLO V3 network architecture.

### 2.2. Anchor Box

The concept of an anchor box is introduced in Faster RCNN, and k-means is used in YOLO V3 to find the anchor box size ratio to detect the target. Instead of directly mapping the coordinates of the bounding box, the parameters relative to the anchor box are predicted. There are nine anchor boxes of YOLO V3 based on the COCO (a large-scale object detection, segmentation, and captioning dataset) dataset; the nine clusters are: $(10 \times 13)$; $(16 \times 30)$; $(33 \times 23)$; $(30 \times 61)$; $(62 \times 45)$; $(59 \times 119)$; $(116 \times 90)$; $(156 \times 198)$ and $(373 \times 326)$. Different size feature maps correspond to different sizes of priori frames.

Therefore, although YOLO V3 can use any reasonable set of anchor boxes for model convergence, the anchor box can be selected in a targeted manner by analyzing the training samples of the input training data set, and we can achieve more effective training convergence. The finer the grid cell can detect finer objects, and the larger the scale, the smaller the receptive field and the more sensitive it is to small objects. Therefore, YOLO V3 assigns the following nine anchors to the prediction output of three different scales. In YOLO V3, the size and distribution of COCO data set's nine anchors are shown in Figure 3.

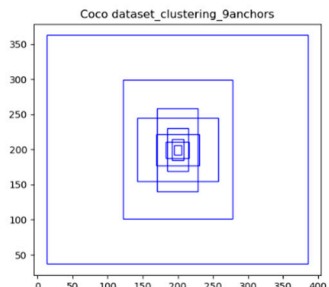

- The anchor boxes allocated by the 13 × 13 scale prediction output layer are (116 × 90), (156 × 198) and (373 × 326).
- The anchor boxes allocated by the 26 × 26 scale prediction output layer are (30 × 61), (62 × 45) and (59 × 119).

- The anchor boxes allocated by the 52 × 52 scale prediction output layer are (10 × 13), (16 × 30) and (33 × 23).

**Figure 3.** Size and distribution diagram of the nine anchors of COCO dataset.

### 2.3. Loss Function

In the deep learning network training, the error between the predicted value and the real value is calculated by the loss function, using the idea of error backpropagation in the neural network, and constantly adjusting the weight of each layer in the network to complete the training of the model. In this process, the loss function determines the direction in which the model is trained. The loss function of YOLO V3 is the weighted sum of the localization loss (errors between the predicted boundary box and the ground truth), the classification loss and the confidence loss (the object of the box), where localization loss attribute (*x*, *y*, *w*, *h*) uses MSE (Mean squared error) and the latter two use cross-entropy.

$$Loss = Error_{localization} + Error_{class} + Error_{confidence} \tag{1}$$

$$Error_{localization} = \sum_{i=1}^{\text{max}box\_number} [(x_{truth} - \hat{x}_{predict})^2 + (y_{truth} - \hat{y}_{predict})^2 + (w_{truth} - \hat{w}_{predict})^2 + (h_{truth} - \hat{h}_{predict})^2] \tag{2}$$

$$Error_{class} = \sum_{i=1}^{\text{max}box\_number} [-class_{truth} \times \log class_{predict} - (1 - class_{truth}) \times \log(1 - class_{predict})] \tag{3}$$

$$Error_{confidence} = \sum_{i=1}^{\text{max}box\_number} [-confidence_{truth} \times \log confidence_{predict} - (1 - confidence_{truth}) \times \log(1 - confidence_{predict})] \tag{4}$$

## 3. Methodologies

To realize the detection target of PCB electronic components based on machine vision, we should not only identify the electronic components by category but also locate the device location. According to the previous review of target detection technologies, we selected YOLO V3, which has both accuracy and timeliness, as the basic model. We decided to implement algorithm improvement from the following aspects:

- Establishment of image database: PCB electronic component image database is the foundation of object detection based on deep learning. The training database is established by combining the real data with the virtual data;
- Size analysis of detection targets: The slicing technique is adopted to separate the labeled image database training targets into different pictures, and the category and size database of detection targets is established through statistical analysis of this picture's size and quantity;
- Network structure improvement: We will analyze the output data content of the detection layer of YOLO V3, determine the feature fusion method according to the detection target and design the network structure improvement scheme;
- Anchor box generated by cluster analysis: A new anchor box is generated by clustering according to PCB electronic components training database to predict bounding box.

### 3.1. Data Acquisition

#### 3.1.1. Label Data Pre-Processing

As the most basic unit of 3C manufacturing, electronic components can be found according to the statistics of website findchips [20]. From the web we can see there are 35 categories, and each category has 2000 or more devices. We can say electronic components come in a wide variety of shapes and sizes. Therefore, no universal data set is currently available. At the same time, electronic components on PCB can be used for special professional devices; yet, the labeling work must be experienced, and experienced professionals can take time and effort to complete their work.

To solve the problem of dataset establishment, we used 47 photos of the physical photos published in [15] as data sets. Some of these PCB photos came from the Internet, while others came from an industrial camera. This data set is available in [15]. Before using the pcb_wacv_2019 data set, we did a statistical analysis of the label data of the data set. We found that the number of labels in the original data set was 17,829, the label names were 5285 and the number of text label was 9190. The label names were composed of the device name + silkscreen number, such as 'capacitor C18', 'resistor R2' and so on. The label information in this data set contains all the devices and text content on the PCB and therefore is very comprehensive, but for our detection task—the identification of electronic components—the silkscreen number and text are redundant information. Therefore, we have merged and deleted the label information. We merged the label data with the same device name and deleted the text label. The processed label name and number statistics are shown in Table 1.

**Table 1.** Statistical analysis table of pre-processing label data.

| Label Name | Battery | Button | Buzzer | Capacitor | Clock | Component |
|---|---|---|---|---|---|---|
| Label number | 1 | 85 | 1 | 2534 | 37 | 985 |
| Label name | connector | connector Port | diode | display | electrolytic | emi |
| Label number | 620 | 1 | 72 | 5 | 246 | 51 |
| Label name | ferrite | fuse | heatsink | ic | inductor | jumper |
| Label number | 30 | 7 | 4 | 386 | 69 | 85 |
| Label name | led | pads | pins | potentiometer | resistor | switch |
| Label number | 214 | 332 | 317 | 7 | 2191 | 58 |
| Label name | test | transformer | transistor | unknown | zener | |
| Label number | 292 | 1 | 84 | 281 | 5 | |

emi: Electromagnetic interference; ic: Integrated circuit.

#### 3.1.2. Virtual Image with Synthesized Data

As everyone knows, data is king. The more high-quality data we have, the better our deep learning models perform. Synthetic data [21,22] can meet specific needs or certain conditions that may not be found in the original, real data. Synthetic data is increasingly being used for machine learning applications: a model is trained on a synthetically generated dataset with the intention of transfer

learning to real data. Efforts have been made to construct general-purpose synthetic data generators to enable data science experiments [23]. In general, synthetic data has several natural advantages:

- Once the synthetic environment is ready, it is fast and cheap to produce as much data as needed;
- Synthetic data can have perfectly accurate labels, including labeling that may be very expensive or impossible to obtain by hand;
- The synthetic environment can be modified to improve the model and training;
- Synthetic data can be used as a substitute for certain real data segments that contain for example, sensitive information.

As mentioned earlier, there are many kinds of electronic components and various shapes. It is almost impossible to collect real photos of PCBs containing most electronic components to build a training dataset. At the same time, even experienced engineers are unlikely to identify all device categories for labeling. For these reasons, we used the Altium Designer which provided Octo part, a component search engine that accesses hundreds of distributors, thousands of manufacturers and millions of electronic components. This enables us to take advantage of prototypes in the prototype common parts library to generate PCB virtual images with electronic components. These virtual images are visually almost identical to real boards. The synthesized data of the virtual image can directly query the parts name and category, and the deep learning engineer, who has no professional electrical knowledge, can also directly name the label according to the query result. The combination of synthetic data and real data to form a training data set can reduce the cost of data collection and improve the generalization of the training model [24]. The display and query process of the PCB virtual image in the Altium Designer is shown in Figure 4.

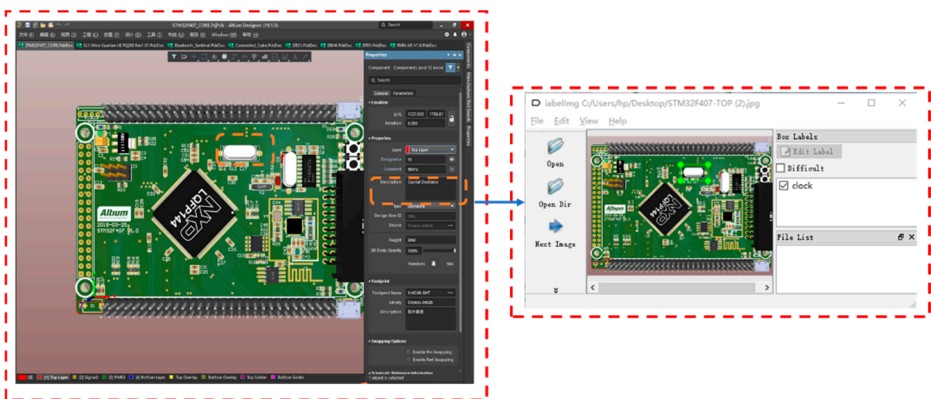

**Figure 4.** Display query and label of virtual image with synthesized data.

### 3.2. Size Analysis of Detection Targets

PCB electronic components are mainly divided into two types—SMD and PTH (Plated through holes)—due to different assembly processes. SMD devices tend to be smaller in size and larger in PTH size. Taking ML365_Top as an example, by slicing the labeled target on the training picture, we show the original image, the name and size of the different types of label components (Table 2) and can observe that the size difference of different types of devices is large. To adapt to the different feature scales of different depth convolution layers of YOLO V3, we will carry out dimensional statistical analysis on the devices in the training set to understand the size distribution of different types of devices.

**Table 2.** Statistical analysis table of PCB-labeled electronic components image size.

| Image of ML365_Top | Labeled Components | |
|---|---|---|
|  | Button 18<br>148 × 147 | Display 0<br>661 × 1466 |
| | Ic 159<br>140 × 182 | Connector 248<br>170 × 173 |
| | led49<br>49 × 27 | Switch 6<br>193 × 192 |
| | Resistor 1096<br>30 × 13 | Capacitor 1198<br>60 × 30 |
| | Inductor 25<br>58 × 29 | Potentiometer 0<br>191 × 188 |

## 3.3. Network Structure Improvement

YOLO V3 uses 53-layer Darknet-53 as the feature extractor. The two most important parts of Darknet-53 are convolutional and resnet. Here, the $1 \times 1$ convolution can compress the number of feature map channels to reduce the model calculation and parameters. Multiple $3 \times 3$ convolution layers are more nonlinear than a large filter convolution layer, making the decision function more decisive. Resnet can make the network deeper, faster, easier to optimize, with fewer parameters and lower complexity than previous models. Therefore, it can solve the problem of deep network degradation and training difficulty. Darknet-53 performs a total of five dimensionality reduction operations. The number of rows and columns which belong to the feature matrix of each dimension reduction output becomes half while the depth doubles compared with the previous. Figure 5 shows the corresponding feature output of different reduction dimension modules of the entire Darknet-53.

Through the Darknet-53 feature output diagram, we can see that each pixel on the feature map of different layers represents the different sizes of the original image. For example, each pixel in the image $104 \times 104 \times 128$ represents the size $4 \times 4$ of the original 416× 416 image. At the same time, each pixel in the image $13 \times 13 \times 1023$ represents the size $32 \times 32$ of the original $416 \times 416$ image. For this reason, YOLO V3 chose $52 \times 52 \times 256$, $26 \times 26 \times 512$ and $13 \times 13 \times 1024$, three feature output layers for upsampling and feature fusion to detect large, medium and small objects of different sizes. As we found in the previous statistical analysis, in the PCB electronic component identification task, the number of SMD much like resistor, capacitor and LED (Light-emitting diode), is large, while the size is small. Each training picture has different resolutions and different sizes due to different acquisition methods. Before training, YOLO V3 resizes the size of pictures to $416 \times 416$. This is often done for the label content in the original image with compression processing and the detection target of a small size will become smaller. If only the deep feature output of Darknet-53 is selected for training and learning, the small size components will lose feature information. Therefore, to improve the accuracy of target detection, we must add a shallow layer feature output to the component category judgment and positioning based on the original YOLO layer. The improved YOLO V3 network structure is shown in Figure 6.

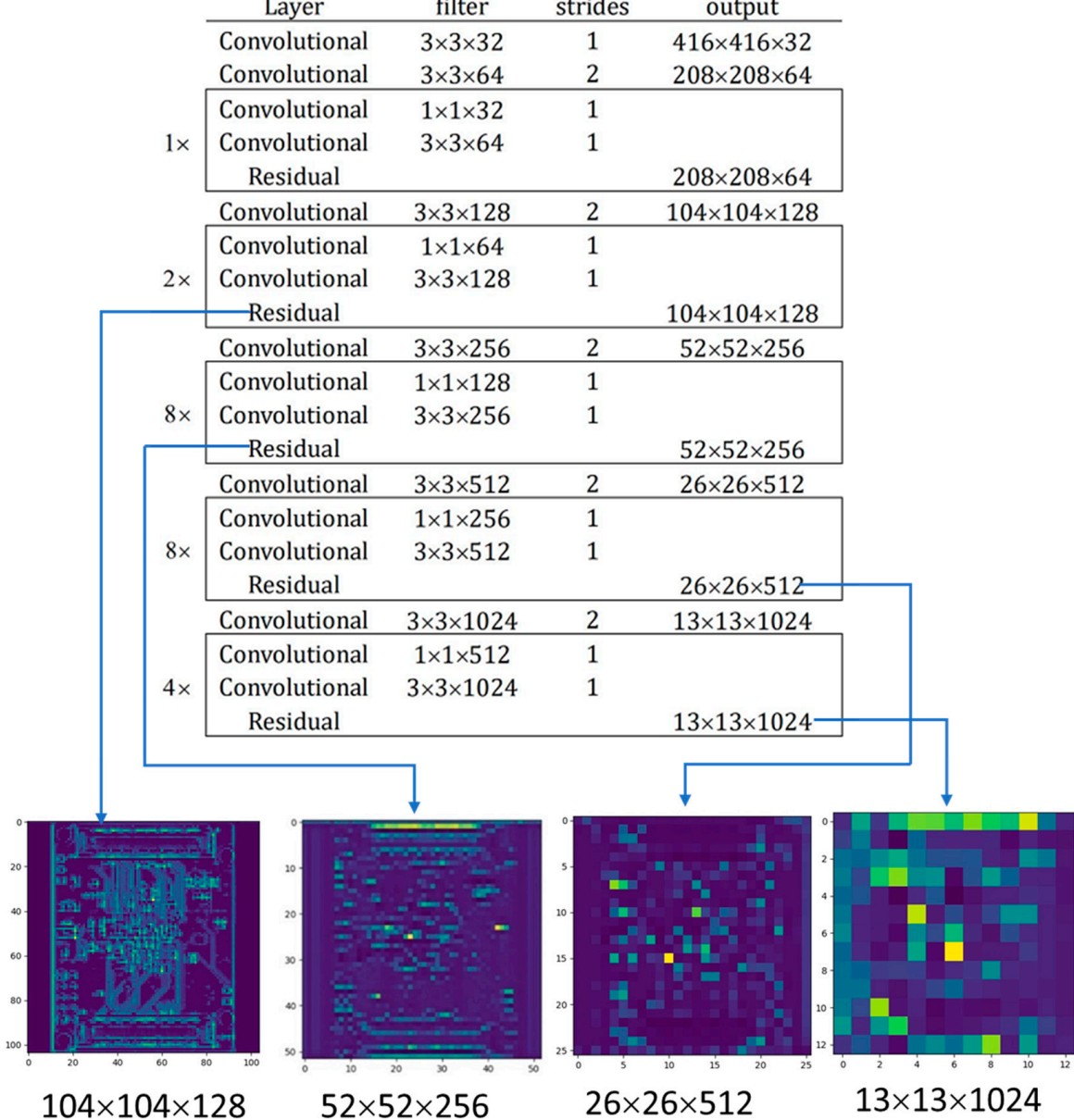

**Figure 5.** Darknet-53 feature output schematic.

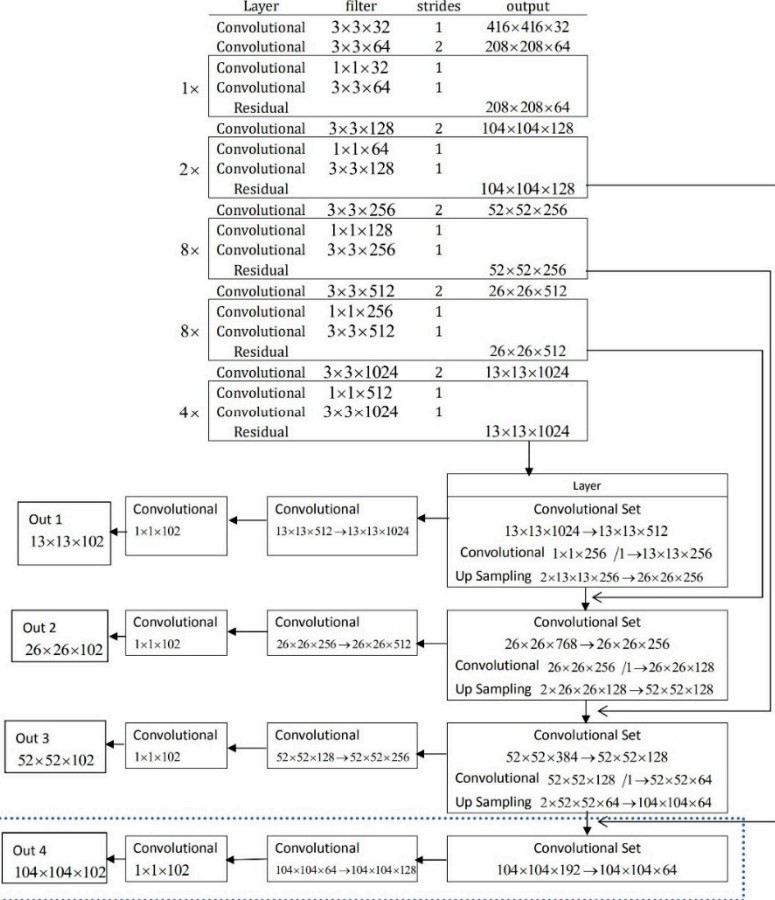

**Figure 6.** Improved network structure of YOLO V3.

### 3.4. K-means++ Clustering Generates Anchor Box

YOLO V3 uses the concept of anchor boxes when doing bounding box prediction. The meaning of an anchor box is the most likely the object width and height. This is obtained by clustering in advance; for example, when we want to use a pixel to predict an object. Around this pixel, we can predict the shape of an infinite number of objects. It is not a random prediction. It is necessary to find the size of the anchor box that is the most likely to be counted from the labeled data. In Section 2, we showed the schematic diagram of the nine anchor box sizes generated by the original YOLO V3 based on the COCO data set. In order to adapt to the detection target of PCB electronic components, we use the k-means++ algorithm to generate nine and 12 anchor boxes according to the three output ports of original YOLO V3 and the four output ports of the improved YOLO V3. We can get nine anchor boxes of YOLO V3 based on the PCB dataset through the calculation; the nine clusters are: $(13 \times 31)$; $(21 \times 42)$; $(31 \times 15)$; $(34 \times 58)$; $(51 \times 29)$; $(57 \times 98)$; $(78 \times 48)$; $(150 \times 118)$ and $(255 \times 323)$. There are 12 anchor boxes of improved YOLO V3 based on the PCB dataset; the 12 clusters are: $(13 \times 24)$; $(14 \times 34)$; $(19 \times 10)$; $(24 \times 14)$; $(28 \times 54)$; $(33 \times 15)$; $(35 \times 33)$; $(47 \times 23)$; $(54 \times 87)$; $(69 \times 45)$; $(146 \times 118)$ and $(255 \times 323)$. We assign these anchor boxes to the improved YOLO V3. The anchor boxes allocated by the $13 \times 13$ scale prediction output layers are $(69 \times 45)$, $(146 \times 118)$ and $(255 \times 323)$; the $26 \times 26$ scale prediction output layers are $(35 \times 33)$, $(47 \times 23)$ and $(54 \times 87)$; the $52 \times 52$ scale prediction output layers are $(24 \times 14)$, $(28 \times 54)$ and $(33 \times 15)$; and the $104 \times 104$ scale prediction output layers are $(13 \times 24)$, $(14 \times 34)$ and $(19 \times 10)$. Figure 7 shows nine cluster anchor boxes and 12 cluster anchor boxes generated based on the PCB dataset.

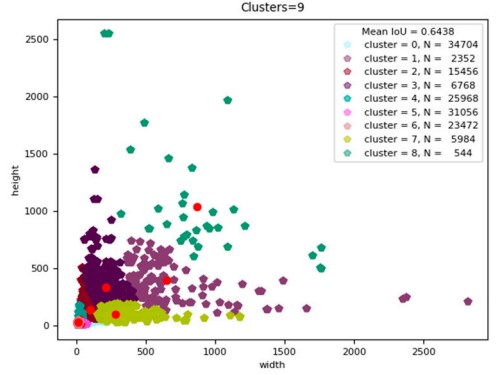

(**a**) Clusters = 9 schematic diagrams on PCB dataset.

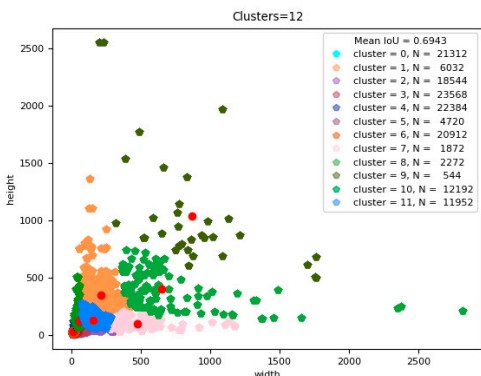

(**b**) Clusters = 12 schematic diagrams on PCB dataset

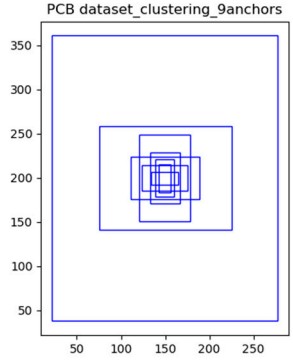

(**c**) Nine anchor boxes on PCB dataset.

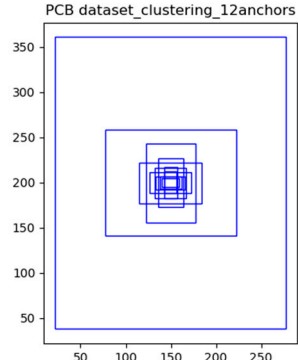

(**d**) Twelve anchor boxes on PCB dataset.

**Figure 7.** K-means++ clustering and generating anchor.

## 4. Experiments and Analysis

To test our proposed method, we have reorganized resources to establish a new dataset of PCB electronic components based on network and virtual images. The PCB dataset contains 50 images in total. Although the proposed method is extremely flexible, we instantiate one for the experiments and provide implementation details. Furthermore, a series of experiments and ablation studies are conducted to show the effectiveness of each design choice.

### 4.1. Image Data Augmentation

Image data augmentation [25] is a technique that can be used to artificially expand the size of a training dataset by creating modified versions of images in the dataset. Training deep learning neural network models on more data can result in more skillful models, and the augmentation techniques can create variations of the images that can improve the ability of the fit models to generalize what they have learned to new images.

The dataset used in the experiment is a joint dataset. It contains the pre-processed PCB dataset and virtual dataset mentioned earlier. There are 50 images, 29 instruments categories and 9145 electronic components in the dataset. Compared with the electronic components in the template image in different positions, poses, scales, illumination, noise, etc., the components in the template image can be translated, rotated, scaled, added, brightened, darkened, etc. Data augmentation expands the current dataset to 20 times the original dataset. The completed augmentation dataset is divided into 8:2, that is, 8 pieces of data are randomly selected for training, and 2 pieces of data are used as detection data. The number and category of the electronic component after image data augmentation is shown in Figure 8.

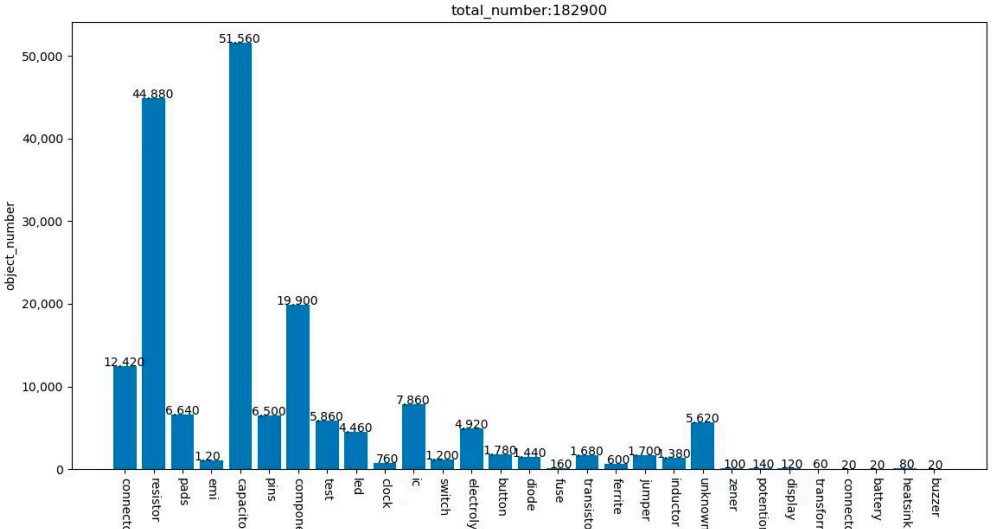

**Figure 8.** Number and category of electronic components after data augmentation.

## 4.2. Implementation Detail

### 4.2.1. Experimental Platform

The experimental platform is the operating system (OS): Windows 10; core processor (CPU): Intel Xeon 6132 × 22.60 GHz; graphics processor (GPU): NVIDIA Quadro P5000; hard disk space: 512GSSD+2T SATA; memory: 192 GB, Python 3.5.2.

### 4.2.2. Experimental Parameter Setting

As we know, we usually use a bounding box to describe the target location in object detection. The bounding box is a rectangular box that can be determined by identifying the x and y coordinates of the upper left corner and the height and width of the box (bx, by, bh, bw). The input image is divided into an S × S (in this paper, S × S means 13 × 13, 26 × 26, 52 × 52 and 104 × 104) grid of cells. For each object that is present on the image, one grid cell is said to be "responsible" for predicting it. Each grid cell predicts three bounding boxes as well as 29 category probabilities. Electronic components on the PCB are closely distributed and numerous. To accurately identify the electronic components, we analyzed the number of electronic components on the PCB, as shown in Figure 9. Therefore, it was determined that the total number of predicted bounding boxes on each test image is not less than 800. Therefore, the original YOLO V3 bounding boxes with 120, and the improved 800, are tested respectively in the experiments.

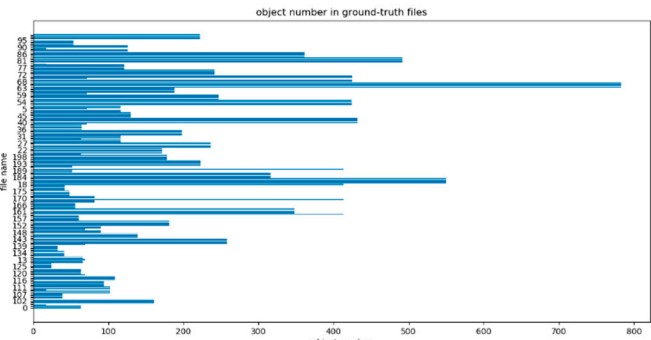

**Figure 9.** Quantity analysis chart of test picture electronic components.

The four considered convolutional networks, YOLO V3+COCO dataset 9 anchors, YOLO V3 + PCB dataset 9 anchors, YOLO V3 + PCB dataset 9 anchors 800 bounding boxes and YOLO V3 + 4 outputs PCB dataset 12 anchors 800 bounding boxes, are implemented respectively in TensorFlow 1.14. Table 3 shows the parameters used for the four experimental algorithms.

**Table 3.** Experimental execution parameters.

| Algorithm | Parameter | Value |
|---|---|---|
| All four convolutional networks follow | Train image size in pixels (height × width) | 416 × 416 |
| | Number of categories | 29 |
| | Training steps | 70,000 |
| | Learn_rate_init | 0.0001 |
| | Learn_rate_end | 0.000001 |
| | Weight decay | 0.0005 |
| | Gradient Descent | Adam Optimizer |
| | Train mode | GPU (Graphics Processing Unit) |
| YOLO V3 + COCO dataset 9 anchors | Self.max_bbox_per_scale | 120 |
| | Anchors | (10 × 13); (16 × 30); (33 × 23); (30 × 61); (62 × 45); (59 × 119); (116 × 90); (156 × 198); (373 × 326) |
| | Outputs | 3 |
| YOLO V3 + PCB dataset 9 anchors | Self.max_bbox_per_scale | 120 |
| | Anchors | (13 × 31); (21 × 42); (31 × 15); (34 × 58); (51 × 29); (57 × 98); (78 × 48); (150 × 118); (255 × 323) |
| | Outputs | 3 |
| YOLO V3 + PCB dataset 9 anchors 800 boarding boxes | Self.max_bbox_per_scale | 800 |
| | Anchors | (13 × 31); (21 × 42); (31 × 15); (34 × 58); (51 × 29); (57 × 98); (78 × 48); (150 × 118); (255 × 323) |
| | Outputs | 3 |
| YOLO V3 + 4 outputs PCB dataset 12 anchors 800 boarding boxes | Self.max_bbox_per_scale | 800 |
| | Anchors | (13 × 24); (14 × 34); (19 × 10); (24 × 14); (28 × 54); (33 × 15); (35 × 33); (47 × 23); (54 × 87); (69 × 45); (146 × 118); (255 × 323) |
| | Outputs | 4 |

*4.3. The Experimental Results*

To better illustrate the effectiveness of the proposed algorithm, we will illustrate by showing the detection image results, the accuracy of the table of detection, and a series of curves of the four algorithms.

4.3.1. Analysis of Subjective Test Results

The above four algorithms can realize multi-object detection in one image and we tested 200 images with them. The detection object is matched in different bounding boxes and some detection results are expressed as follows (Figures 10–12). The four subfigures a–d of Figures 10–12 represent the detection effects of the above four algorithm experiments. For the sake of simplicity, we use algorithms in subfigures a–d of Figures 10–12 to represent YOLO V3 + COCO dataset 9 anchors, YOLO V3 + PCB dataset 9 anchors, YOLO V3 + PCB dataset 9 anchors 800 boarding boxes, and YOLO V3 + 4 outputs PCB dataset 12 anchors 800 boarding boxes from Figures 10–12. The algorithm in every subfigure d of Figures 10–12 is our final improved YOLO V3 method for detecting PCB electronic components. Here we use yellow lines to mark the difference between the four algorithms.

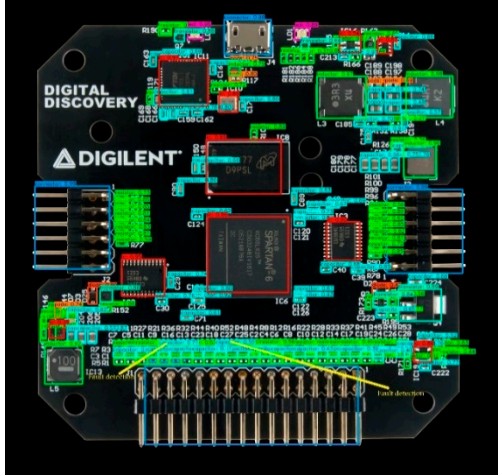

(**a**) YOLO V3 + COCO dataset 9 anchors.

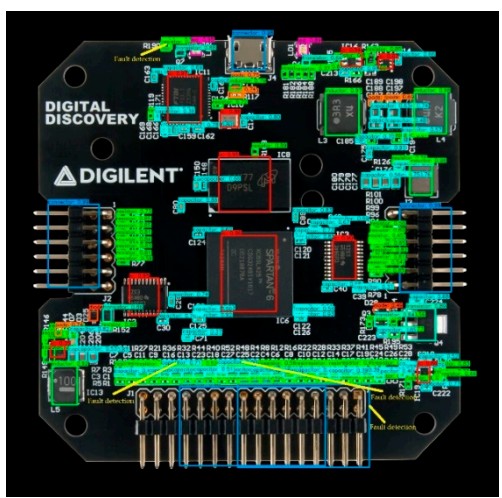

(**b**) YOLO V3 + PCB dataset 9 anchors.

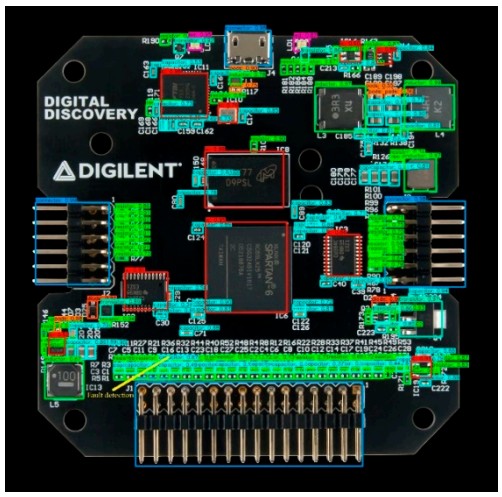

(c) YOLO V3 + PCB dataset 9 anchors 800 boarding boxes.

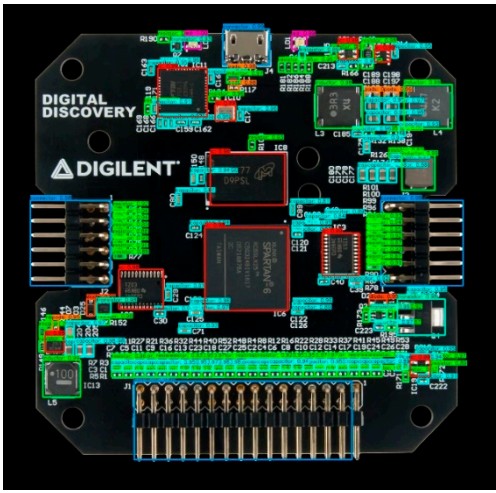

(**d**) YOLO V3 + 4 outputs PCB dataset 12 anchors 800 boarding boxes.

**Figure 10.** Comparisons of small object detection results between the four algorithms.

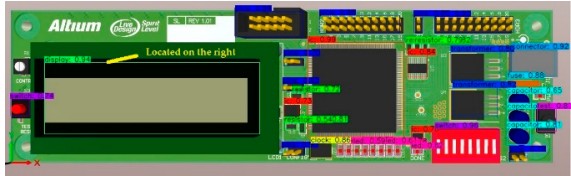

(**a**) YOLO V3 + COCO dataset 9 anchors.

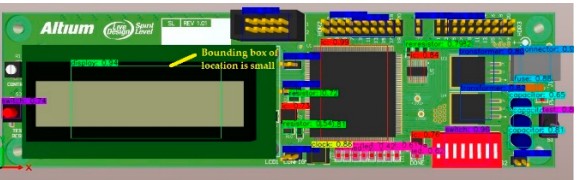

(**b**) YOLO V3 + PCB dataset 9 anchors.

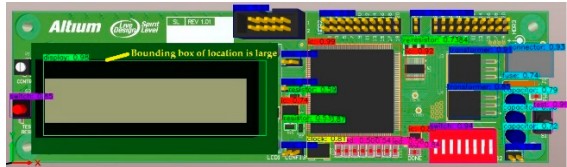

(**c**) YOLO V3 + PCB dataset 9 anchors 800 boarding boxes.

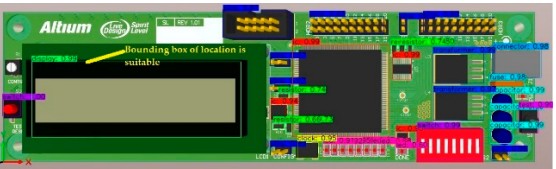

(**d**) YOLO V3 + 4 outputs PCB dataset 12 anchors 800 boarding boxes.

**Figure 11.** Comparisons of object detection results on virtual image between the four algorithms.

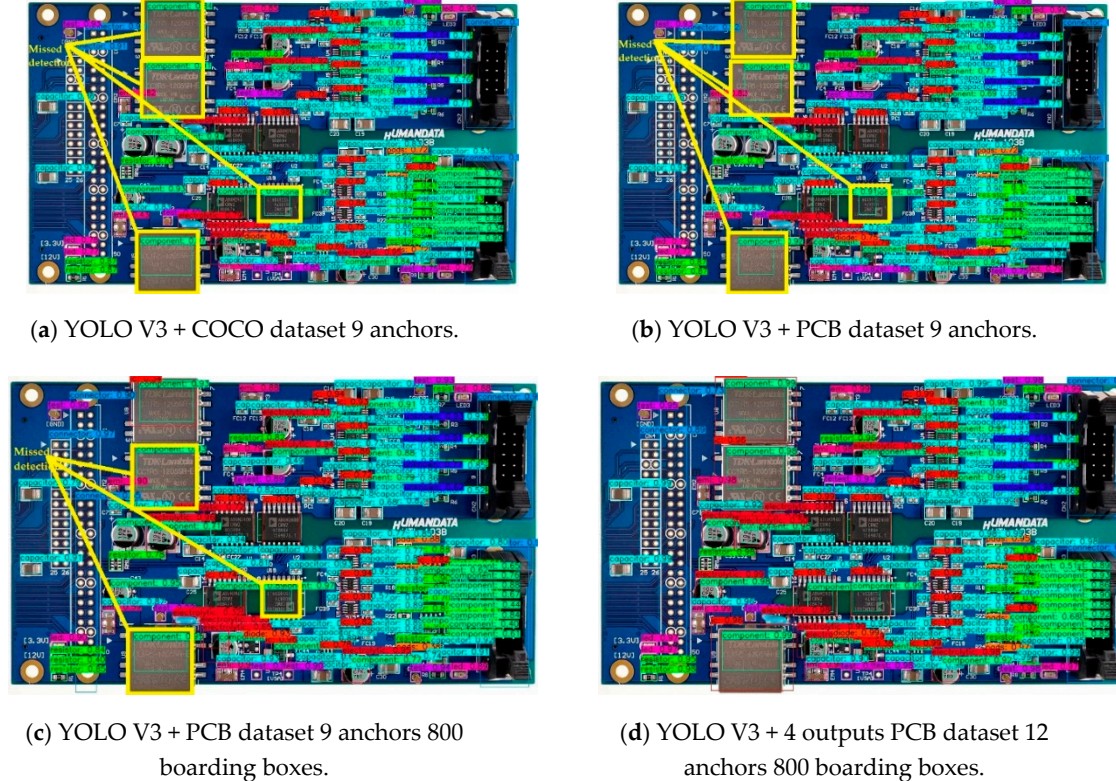

(**a**) YOLO V3 + COCO dataset 9 anchors.

(**b**) YOLO V3 + PCB dataset 9 anchors.

(**c**) YOLO V3 + PCB dataset 9 anchors 800 boarding boxes.

(**d**) YOLO V3 + 4 outputs PCB dataset 12 anchors 800 boarding boxes.

**Figure 12.** Comparisons of large object detection results between the four algorithms.

Figure 10 highlights the effects of the four algorithms on detecting small objects. Small objects on the PCB mainly refer to electronic components such as resistors, capacitors and LEDs that are small in size. Therefore, we focus on the identification of three rows of resistors and capacitors in the lower middle of the picture. From Figure 10a we can observe that there are obvious detection errors in either the resistor or the capacitor. We can see that most of the detected bounding boxes size have changed, but it does not significantly improve the accuracy of the target detection in Figure 10b. The algorithm in Figure 10c considers the number of electronic components to be detected on each PCB picture based on the algorithm in Figure 10b. It is necessary to detect up to 800 targets per picture in the detection algorithm. Therefore, we can see that the detection accuracy of the resistance has been improved in Figure 10c. Figure 10d shows that the final improved algorithm can not only correctly detect the small components such as resistors and capacitors, but also the detected target bounding boxes are more suitable for the actual target in size.

This paper proposes a method for training a real PCB photo and a virtual PCB photo with synthesized data to form a joint dataset. Therefore, we used the training results to detect the electronic components on the PCB virtual picture. Figure a–d of Figure 11 shows the results of the test. We can see that all four algorithms implement correct detection of electronic components, but the algorithm in Figure 11d is more accurate in detecting the position information of components.

Figure 12 shows the effect of the four algorithms on detecting larger size targets. Now we are focusing on IC (integrated circuit) devices on the PCB. From Figure 12a, we can see that the three ICs on the left side have not been detected; this means the original YOLO V3 still has problems in detecting larger-sized devices. Figure 12b does not show significant changes. After increasing the number of bounding boxes in each picture, we can observe that some ICs have been detected in Figure 12c,d which shows that all ICs are displayed in bounding boxes accurately.

### 4.3.2. Analysis of Objective Test Results

For the detection of PCB electronic components containing 29 categories, we used the AP (average precision) detected by each category of components to characterize the performance of the four algorithms. In Table 4, we use bold to indicate that the result of this algorithm is better than or equal to other algorithms. From Table 4 we can learn from the specific data that YOLO V3+4 outputs + PCB dataset 12 anchors + bbox800 have improved detection accuracy in all categories.

**Table 4.** AP for each electronic component category of four algorithms.

| Category | AP (Average Precision) of Algorithm | | | |
|---|---|---|---|---|
| | YOLO V3 + COCO Dataset 9 Anchors | YOLO V3 + PCB Dataset 9 Anchors | YOLO V3 + PCB Dataset 9 Anchors + Bbox800 | YOLO V3 + 4 Outputs+ PCB Dataset 12 Anchors + Bbox800 |
| resistor | 0.41 | 0.41 | 0.38 | **0.48** |
| capacitor | 0.58 | 0.58 | 0.56 | **0.72** |
| test | 0.82 | 0.81 | 0.81 | **0.96** |
| unknown | 0.74 | 0.74 | 0.72 | **0.87** |
| emi | 0.91 | 0.93 | 0.94 | **0.99** |
| ferrite | 0.78 | 0.75 | 0.75 | **0.88** |
| pads | 0.55 | 0.53 | 0.55 | **0.74** |
| led | 0.88 | 0.89 | 0.87 | **0.94** |
| zener | 1 | 1 | 0.80 | **1** |
| component | 0.49 | 0.50 | 0.52 | **0.56** |
| transistor | 0.88 | 0.92 | 0.85 | **0.99** |
| diode | 0.80 | 0.83 | 0.82 | **0.94** |
| jumper | 0.71 | 0.77 | 0.75 | **1** |
| inductor | 0.74 | 0.75 | 0.62 | **0.96** |
| fuse | 0.59 | 0.51 | 0.44 | **1** |
| electrolytic | 0.67 | 0.65 | 0.53 | **0.99** |
| transformer | 1 | 1 | 1 | **1** |
| potentiometer | 0.71 | 0.54 | 0.61 | **1** |
| pins | 0.91 | 0.93 | 0.93 | **0.98** |
| clock | 0.60 | 0.61 | 0.55 | **1** |
| battery | 0.12 | 1 | 0.75 | **1** |
| button | 0.98 | 0.99 | 0.99 | **1** |
| ic | 0.74 | 0.77 | 0.75 | **0.98** |
| switch | 0.78 | 0.78 | 0.76 | **1** |
| connector | 0.94 | 0.96 | 0.95 | **1** |
| connector port | 1 | 1 | 1 | **1** |
| buzzer | 1 | 1 | 1 | **1** |
| heatsink | 1 | 1 | 1 | **1** |
| display | 1 | 1 | 1 | **1** |

The comparison of objective analyses shows that the detection effect of the algorithm in every subfigure d of Figures 10–12 is better than the other three detection models. This result is mainly due to the structural improvement of the YOLO V3 neural network, the increase of the output's number, and the improvement of all components' detection network accuracy.

### 4.3.3. Analysis of a Series of Curves

To comprehensively compare and analyze the advantages and disadvantages of the four algorithms, we have drawn four curves for comparative analysis in Figure 13. Among the four curves, we use orange to represent YOLO V3 + COCO dataset 9 anchors algorithm, green for YOLO V3 + PCB dataset 9 anchors algorithm, red for YOLO V3 + PCB dataset 9 anchors + bbox800 algorithm, and blue for YOLO V3 + 4 outputs + PCB Dataset 12 anchors + bbox800 algorithm.

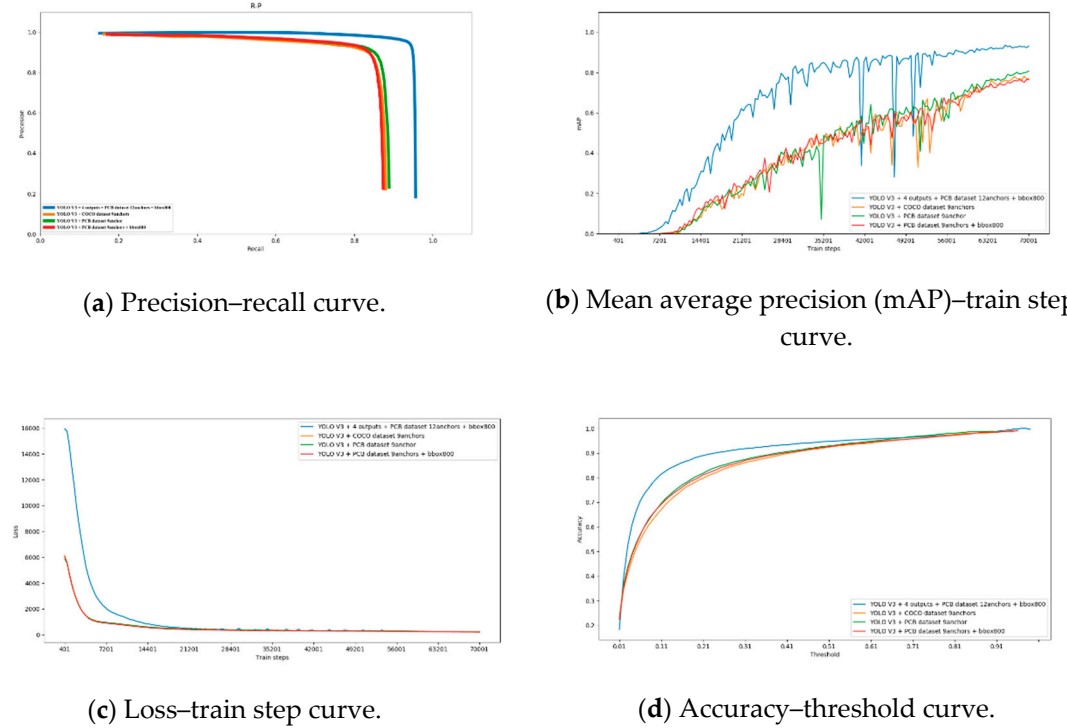

(**a**) Precision–recall curve.

(**b**) Mean average precision (mAP)–train steps curve.

(**c**) Loss–train step curve.

(**d**) Accuracy–threshold curve.

**Figure 13.** Evaluation curves of the four algorithms.

Precision recall is a useful measure of success of prediction when categories are very imbalanced. In information retrieval, precision is a measure of result relevancy, while recall is a measure of how many truly relevant results are returned. With the accuracy of the *y*-axis and the recall rate of the *x*-axis, we get the precision–recall (P-R) curve. The P-R curve shows the tradeoff between precision and recall for different thresholds. The higher the accuracy, the higher the recall rate and our models and algorithms are more efficient. That is, the drawn P-R curve is as close as possible to the upper right. As shown in Figure 13a, the blue line is closest to the upper right and encloses the other three algorithm curves. Therefore, the improved algorithm we finally propose is represented by the color blue and shows the best performance of the four algorithms.

The mAP (mean average precision) provides a single-figure measure of quality across recall levels. Among evaluation measures of different target detection algorithms, mAP has been shown to have especially good discrimination and stability. In the experiments of this paper, the training steps have been iterated a total of 70,000 times. We have the mAP value as the *y*-axis and the range is from 0 to 100%; the number of iterations is the *x*-axis, which ranges from 0 to 70,000. We can see from Figure 13b, when the number of training steps reaches 50,000, the best performing blue curve mAP has been stabilized, and the mAP maximum is 93.07%. This is the model selected in this paper.

Figure 13c shows the loss value curve changes with iterations of the four algorithms. It can be seen that the orange, red and green curves correspond to the algorithm quickly fitting in the first 7000 iterations, then the loss gets smaller quickly and then gradually stabilizes after 20,000 iterations. The loss value of the blue curve is higher at the starting point, but it can also reach the same stable value as the other three curves after 21,200 iterations.

Figure 13d shows the relationship between the accuracy and confidence of the target detection. It is known from Figure 13d that as the confidence level increases, the accuracy of target recognition also increases. During detection, the accuracy and confidence provided by the blue line model are significantly higher than the other three models, reflecting the superiority of the YOLO V3 + 4 outputs + PCB dataset 12 anchors yolo v3 + bbox800 algorithm.

Through the above experimental results, the improved model of YOLO V3 + 4 outputs + PCB dataset 12 anchors + bbox800 has achieved good results not only in small devices, large devices and virtual pictures, but also in all types of electronic components.

### 4.3.4. Comparisons of Detection Performance between Four Algorithms and Other Methods

In consideration of the detection results, the above four algorithms for YOLO V3 were chosen for comparison with state-of-the-art methods, including Faster RCNN [26] and SSD [27]. The whole tests were evaluated on a single NVIDIA Quadro P5000 graphics card. The mAP and detection speeds of these different network structures are shown in Table 5.

**Table 5.** mAP and detection speed for some object detection methods on the PCB dataset.

| Method | Faster RCNN | SSD (single-shot multi-box detectors) | YOLO V3 + COCO Dataset 9 Anchors | YOLO V3 + PCB Dataset 9 Anchors | YOLO V3 + PCB Dataset 9 Anchors + Bbox800 | YOLO V3 + 4 Outputs + PCB Dataset 12 Anchors + Bbox800 |
|---|---|---|---|---|---|---|
| mAP (%) | 80.25 | 83.16 | 77.14 | 79.88 | 76.43 | 93.07 |
| Run time (sec/img) | 0.55 | 0.32 | 0.39 | 0.385 | 0.395 | 0.41 |

RCNN: Regions with convolutional neural network features.

As seen from Table 5, regardless of whether difficult objects are included, the mAP of YOLO V3 + 4 outputs + PCB dataset 12 anchors + bbox800 is much higher than others. The detection speed of Faster RCNN is slower than those of the regression-based methods. Therefore, the proposed YOLO V3 + 4 outputs + PCB dataset 12 anchors + bbox800 is an effective method for PCB electronic component detection.

### 5. Conclusions

With the improvement of the manufacturing process, materials and technology of electronic components, the appearance of electronic components has the characteristics of different shapes and large changes in size. Vision-based detection of electronic components on PCBs is a core issue in the intelligent manufacturing of electronic products. At present, the use of deep learning to detect PCB electronic components is mainly concentrated in several cases with few types and quantities, and the accuracy is not high. Studying how to use efficient and fast target detection methods to detect a large number of electronic components on a PCB can provide more possibilities for defect detection and robot assembly of electronic products.

In this study, we used YOLO V3, which has good detection speed and good detection results, as the basic framework for electronic component detection on PCB. Firstly, it provides a real PCB photo and a virtual PCB photo with synthetic data as a joint dataset. The synthetic data method provides label information for as many electronic components as possible. Secondly, we analyzed the sample size distribution in the training set and studied the output results of five YOLO V3 network structure dimensionality reduction operations. We found that each pixel represents the different sizes of the original image. This gives us the idea of selecting feature layers according to the size of different detection target categories to improve the detection accuracy. Finally, we generated some new anchor boxes based on the clustering results of PCB electronic components in the training set. Based on the above data, we verified through a series of experiments that the final improvement model increased the mAP of the original YOLO V3 from 77.08% to 93.07%.

Although the final improvement algorithm does improve the overall detection accuracy of electronic components on the PCB, we also found that the AP (average precision) of the responder, capacitor and component was 0.41, 0.58 and 0.49, respectively, in the original YOLO V3. The AP after

the improved algorithm was 0.48, 0.72, and 0.56, respectively. That is, the increase in detection accuracy was small. The problem of resistor and capacitor detection is that the appearance is similar and the variance between categories is small. The detection problem of components is that the characters on the IC are different, and the variance within the category is large. It is these findings that allow us to explore further. As Marco Leo et al. proposed, deep learning can also be used to implement scenario understanding so that we can fully consider the context of electronic components on the PCB [28]. We will perform the detection of the resistor and capacitor according to the silkscreen printing on the PCB. The detection of the component can also be realized by the edge detection method in the IC area.

In the process of studying the YOLO V3 model and improving its algorithm, the most unanticipated finding was that the YOLO algorithm completely provides an open structure for various target detection tasks. We can completely detect the target feature based on the original YOLO network by cropping, deepening, lightening, combining and determine the number of output layers. This will provide a large application space for all kinds of real-time detection tasks.

**Author Contributions:** J.L. conducted the literature review with the discussion, correction, and guidance provided by J.G. and Z.H. and at every stage of the process, from the general structure to the specific details. J.W. provided part of the software implementation. All authors have read and approved the final manuscript.

**Funding:** This research was funded by the National Natural Science Foundation of China (NO.51875266).

**Conflicts of Interest:** The authors declare no conflict of interest.

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
