# Peer review of "Application Research of Improved YOLO V3 Algorithm in PCB Electronic Component Detection"

_applsci, doi:10.3390/app9183750_

Round 1
Reviewer 1 Report
The authors didn't make a full review of the existing articles in the selected subject area (PCB Electronic Component Detection) and various applied methods and approaches to recognition and created datasets. For Example:
Wei Li, Chen Jiang, Matthias Breier and Dorit Merhof, Localizing Components on Printed Circuit Boards Using 2D Information
Sanli Tang, Fan He, Xiaolin Huang, Jie Yang, Online PCB Defect Detector On a New PCB Defect Dataset
and etc.
Figure 8 is absolutely non informative.The difference between
(a) Number and category of electronic component before data augmentation
and
(b) Number and category of electronic component after data augmentation
is only in a proportional increase in the number of components.
Figures 10-13 are also not very informative, since the results of 4 different algorithms (a-d) are visually very similar, and there is no highlighting of incorrectly selected components in these figures.
In the section Analysis of experimental results, there are only comparison of 4 different modifications of the recognition algorithm considered in the article, but there is no comparison in accuracy and completeness with other existing methods and approaches.
In general, the work contains elements of scientific novelty and has good perspectives of practical application.
Reviewer 2 Report
This paper deploys YOLO V3 for recognizing electronic components.
A PCB electronic component category training dataset that involves the virtual environment and real-world scenario collaboration is proposed. Image data augmentation has been used to artificially expand the size of a training dataset.
Validation has been proved in a typical multi-category test set.
The K++ clustering algorithm is to regenerate the anchors suitable for the distribution of PCB electronic components based on the size ratio of the target in the training dataset.
the YOLO V3 network architecture has been modified to make it suitable to detect small targets.
4 different experiments were carried out to recognize components on a PCB. in each experiment the YOLO V3 has been used and the number of anchors and predicted bounding boxes have been modified.
The paper is interesting and recognition performance is very encouraging but, in my opinion, authors should demonstrate how other network configurations work on the same test dataset. In other words, they have to convince the reader that the proposed pipeline is actually the best solution and they have to quantify the improvement with respect state fo the art methods.
As a final remark, I would like to suggest authors add a reference to a recent survey paper was more effective Deep learning approaches used for assistive tasks (even in industrial application contexts) have been listed and discussed.
Deep Learning for Assistive Computer Vision
M Leo, A Furnari, GG Medioni, M Trivedi, GM Farinella
Proceedings of the European Conference on Computer Vision (ECCV)
Round 2
Reviewer 2 Report
Paper has been improved with additional experiments that make definitively clear the contributions of the introduced strategies.